# Critical Study Quality Management for the Anti-Seepage System in Macau's Landfill Area

**Zhaobin Li \***, **Waifan Tang**, **Shulun Mak**, **Qingwen Li**, **Haolin Chen** and **Qianqian Hong**

Department of Construction and Quality Management, School of Science and Technology, Hong Kong Metropolitan University, Homantin Kowloon, Hong Kong SAR 999077, China; s1338585@live.hkmu.edu.hk (H.C.)
\* Correspondence: s1285609@live.hkmu.edu.hk

**Abstract:** This paper delves into the Macau landfill's anti-seepage system project quality management, highlighting its environmental significance. We summarize global research and provide an overview of the project, emphasizing the importance of quality control across phases. We focus on monitoring, evaluation, and inspection methods to ensure quality objectives. Our findings contribute to environmental preservation and sustainability.

**Keywords:** quality management; quality control; anti-seepage system; project management

## 1. Introduction

### 1.1. Introduction to the Background and Importance of the Macau Landfill Anti-Seepage System Project

In the ever-evolving landscape of environmental management, effective quality control within landfill anti-seepage system projects is crucial for ensuring environmental protection and sustainable development. This paper provides an in-depth examination of quality management practices in such projects, shedding light on various aspects critical to success. To better facilitate the understanding of the discussed concepts, Figures 1 and 2 offer visual representations of key components in landfill anti-seepage systems. In the following sections, we will delve into a detailed analysis of these figures to elucidate their significance in the context of project quality management.

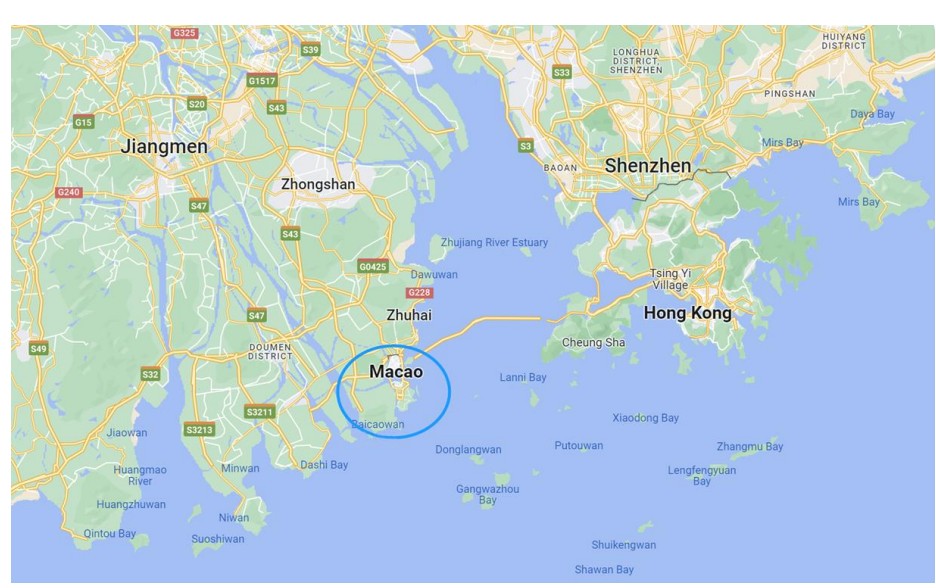

**Figure 1.** Location of Macao.

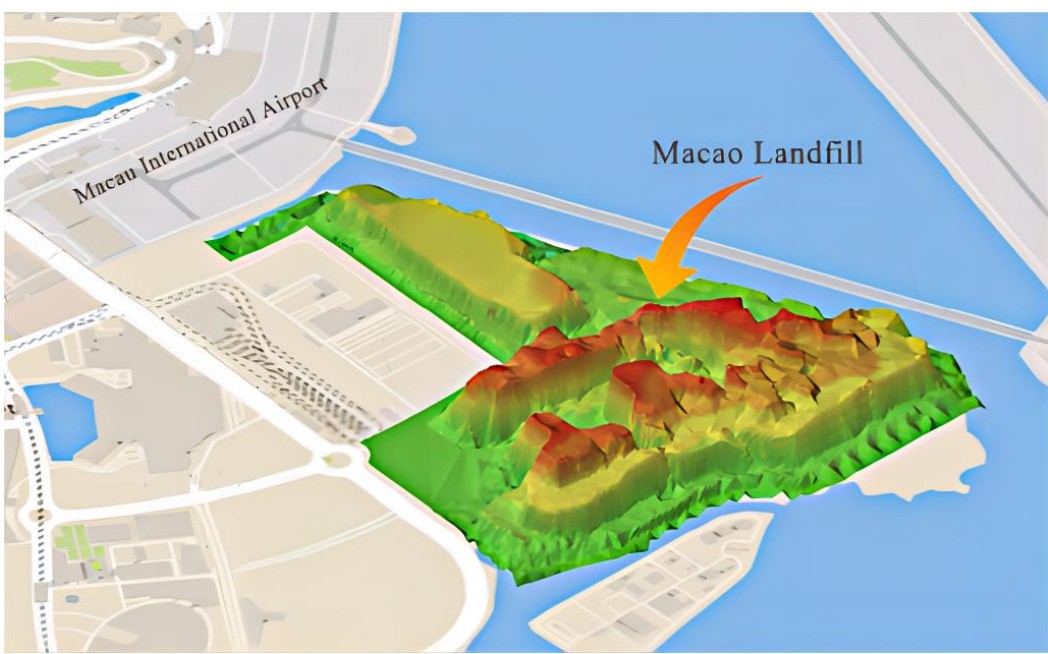

**Figure 2.** Location of Macao landfill.

Geographical location of Macau (Figure 1): Macau is located in the southern part of China, with geographical coordinates of approximately 22.1987° N latitude and 113.5439° E longitude, situated on the western bank of the Pearl River Delta.

Neighboring areas: Macau is adjacent to Zhuhai city and is connected to Guangdong Province, located in the Pearl River Delta region.

Terrain features: It mainly consists of the Macau Peninsula and two smaller islands, Taipa and Coloane.

Location of Macau's landfills (Figure 2):

Location: The landfills in Macau are typically situated on the Macau Peninsula or the nearby islands of Taipa and Coloane.

Geographical layout: Due to limited land resources, landfills might be placed in relatively remote or less developed areas, considering the sustainability of the environment and the community.

Environmental impact: The choice of location for landfills in Macau is related to the locations' geographical characteristics and environmental science, requiring a balance between urban development needs and environmental protection.

As a special administrative region, Macau's geographical limitations make its land resources particularly valuable. Against the backdrop of urban development and population growth, proper waste management has become increasingly vital. Landfills, being a primary waste disposal method, aim to bury waste materials, but they also bring forth potential issues of environmental pollution and ecological degradation. The significance of anti-seepage system projects in landfill construction has become more pronounced, as they effectively reduce the environmental risks posed by waste leakage, safeguarding soil, and groundwater quality.

The anti-seepage system project in the Macau region is designed to ensure environmental safety during waste burial. However, due to variations in geography, climate, and geological conditions, the project faces unique challenges. Therefore, research into quality management for this project is particularly crucial. By delving into quality management methods for anti-seepage system projects, scientific guidance can be provided for project implementation, ensuring environmental health and sustainable development [1].

*1.2. Elaborating the Role of Anti-Seepage Systems in Environmental Protection and Sustainable Development*

Anti-seepage systems play a pivotal role in the field of waste management. Firstly, these systems prevent harmful substances within waste from infiltrating the soil and groundwater, thereby averting groundwater pollution and supporting the sustainable utilization of local water resources. Secondly, by minimizing leakage, adverse impacts on the surrounding ecological environment are mitigated, contributing to ecological balance [2].

Based on data from the Macau construction waste landfill, different types of construction waste, such as inert dismantled materials, sea mud, mixed construction waste, and slag, may possess varying physical properties. Some types of waste may pose greater challenges to the landfill's anti-seepage system because they may be more prone to leakage or affect the efficiency of the anti-seepage layer. Additionally, sea mud is a specialized type of waste that typically requires specific handling and management. If a large amount of sea mud is dumped into the landfill, additional anti-seepage measures may be necessary to ensure there is no contamination of the surrounding soil and groundwater (Figure 3).

**Figure 3.** Source: Macau Environmental Protection Bureau/Total construction waste.

From the data, it can be observed that there is significant fluctuation in the total construction waste volume across different years, with notable peaks in 2014, 2015, and 2020. However, overall, the fluctuations are quite apparent. The handling of construction waste may impact the landfill's anti-seepage system, particularly concerning the types and quantities of waste. To ensure environmental protection and the effectiveness of the anti-seepage system, close monitoring and management of the construction waste disposal process may be required, along with the implementation of appropriate measures to address potential issues (Table 1).

**Table 1.** Source: Macau Environmental Protection Bureau/Total construction waste.

| Year | Total Construction Waste (Cubic Meter) | Inert Dismantled Materials (Cubic Meter) | Sea Mud (Cubic Meters) | Mixed Construction Waste (Cubic Meters) | Slag (Cubic Meters) |
|------|-----------|-----------|-----------|-----------|-----------|
| 2010 | 1,953,821 | 1,267,569 | 371,536   | 271,088   | 43,628  |
| 2011 | 1,617,836 | 1,007,825 | 315,926   | 252,744   | 41,341  |
| 2012 | 2,420,041 | 1,057,979 | 1,096,303 | 215,644   | 50,115  |
| 2013 | 3,925,068 | 2,025,569 | 1,616,290 | 226,308   | 56,901  |
| 2014 | 4,376,182 | 2,327,819 | 1,708,785 | 269,638   | 69,940  |
| 2015 | 4,834,508 | 1,360,528 | 3,098,966 | 270,182   | 104,832 |
| 2016 | 3,269,101 | 928,382   | 1,964,560 | 272,484   | 103,675 |
| 2017 | 3,023,622 | 1,213,095 | 1,407,575 | 312082    | 90,870  |
| 2018 | 2,095,824 | 1,335,009 | 331,227   | 343,944   | 85,644  |
| 2019 | 2,485,693 | 1,642,316 | 420,173   | 331,294   | 91,910  |
| 2020 | 4,063,306 | 1,883,037 | 1,765,400 | 325,078   | 89,791  |
| 2021 | 2,874,736 | 1,325,103 | 1,278,745 | 184,711   | 86,177  |
| 2022 | 2,417,993 | 1,502,722 | 735,254   | 92,384    | 87,633  |

According to the data in Table 1, we can analyze several issues:

a.   Fluctuation in total construction waste: From 2010 to 2022, there is an overall upward trend in total construction waste. Particularly, a sharp increase is observed between 2013 and 2015, suggesting a possible surge in construction activities or demolitions during that period. However, after 2016, the total amount decreases, with a peak again in 2020. This fluctuation may be related to economic activities in the construction industry, policy changes, or other external factors, such as natural disasters;

b.   Changes in proportions of different types of construction waste: The proportions of different types of construction waste (inert dismantled materials, sea mud, mixed construction waste, slag) vary in the total amount and fluctuate over time. For instance, the proportion of inert dismantled materials in total construction waste is constantly changing, reflecting potential shifts in construction material usage and methods. Similarly, the quantities of sea mud and mixed construction waste show significant differences in different years;

c.   Dramatic fluctuation in sea mud quantity: The quantity of sea mud peaks in 2015 and rapidly declines afterward. This may be related to specific construction projects or environmental policies;

d.   Yearly reduction in mixed construction waste: Starting from 2016, the quantity of mixed construction waste decreases annually, indicating an improvement in the efficiency of construction waste classification and processing;

e.   Stable trend in slag production: Compared to other types of construction waste, slag production remains relatively stable with minimal fluctuations.

These data indicate significant variations in both the quantity and types of construction waste over different years, likely influenced by factors such as construction industry activities, policy adjustments, technological advancements, and increasing environmental awareness. These changes also suggest the need for further strategies in construction waste management and recycling.

Based on the data in Table 1, we can propose some relevant theories and considerations for quality management of landfill construction and impermeable systems:

a.   Building waste types and material selection for landfills: The data show variations in the quantity of different types of construction waste, impacting the selection of materials for landfills. For example, the composition and stability of inert disman-

        tled materials and mixed construction waste determine their suitability as landfill materials and the need for specific impermeable measures;

b.    Special considerations for sea mud: Due to its unique properties (high moisture content, high fluidity), special attention is needed when using sea mud in landfills to mitigate its impact on impermeable systems. Sea mud may require special treatment or mixing with other materials to ensure the effectiveness of impermeable systems;

c.    Design and material selection for impermeable layers: The design of impermeable systems should be based on the characteristics of the materials being landfilled. Considerations should include the chemical properties, stability, and permeability of different construction wastes to choose suitable materials and structures for impermeable layers;

d.    Monitoring and management: Establishing an effective monitoring system to continuously monitor leakage in landfills is crucial. This includes regular inspections of the integrity and functionality of impermeable layers, as well as monitoring potential pollution of groundwater and the surrounding environment.

The trend analysis from the data in Table 2 and Figure 4 indicates that there is a noticeable difference between the data for the years 2020 and 2023 when compared annually. Although the data fluctuate overall each year, there is no clear decreasing trend. If the continuous increase in heavy metals is attributed to changes in the composition of waste in the landfill, it may be necessary to consider an increased risk of leakage from the landfill. The higher the concentration of heavy metals in the landfill, the greater the potential impact on the surrounding environment during a leakage event. The accumulation of heavy metals may affect the integrity of the landfill liner, thereby increasing the risk of leakage.

**Table 2.** Source: Macau Environmental Protection Bureau/Received heavy metals.

| Date | Received Heavy Metals/Tonne |
|---|---|
| Jul-23 | 50,817.32 |
| Jun-23 | 47,632.67 |
| May-23 | 49,073.93 |
| Apr-23 | 44,609.88 |
| Mar-23 | 45,382.71 |
| Feb-23 | 40,428.64 |
| Jan-23 | 41,607.79 |
| Dec-22 | 40,618.01 |
| Nov-22 | 43,315.88 |
| Oct-22 | 43,008.89 |
| Sep-22 | 43,934.06 |
| Aug-22 | 42,723.89 |
| Jul-22 | 34,550.75 |
| Jun-22 | 42,297.18 |
| May-22 | 44,462.98 |
| Apr-22 | 41,511.53 |
| Mar-22 | 44,543.56 |
| Feb-22 | 36,951.66 |
| Jan-22 | 48,480.95 |
| Dec-21 | 43,627.71 |
| Nov-21 | 42,563.15 |

**Table 2.** *Cont.*

| Date | Received Heavy Metals/Tonne |
| --- | --- |
| Oct-21 | 41,098.75 |
| Sep-21 | 43,383.92 |
| Aug-21 | 43,117.45 |
| Jul-21 | 44,311.81 |
| Jun-21 | 44,184.58 |
| May-21 | 46,334.50 |
| Apr-21 | 43,449.80 |
| Mar-21 | 44,846.84 |
| Feb-21 | 37,207.11 |
| Jan-21 | 44,155.58 |
| Dec-20 | 47,263.25 |
| Nov-20 | 43,542.26 |
| Oct-20 | 43,195.08 |
| Sep-20 | 44,124.60 |
| Aug-20 | 44,458.37 |
| Jul-20 | 43,145.58 |
| Jun-20 | 46,733.62 |
| May-20 | 40,505.04 |
| Apr-20 | 38,882.11 |
| Mar-20 | 38,682.82 |
| Feb-20 | 30,461.47 |
| Jan-20 | 47,311.29 |

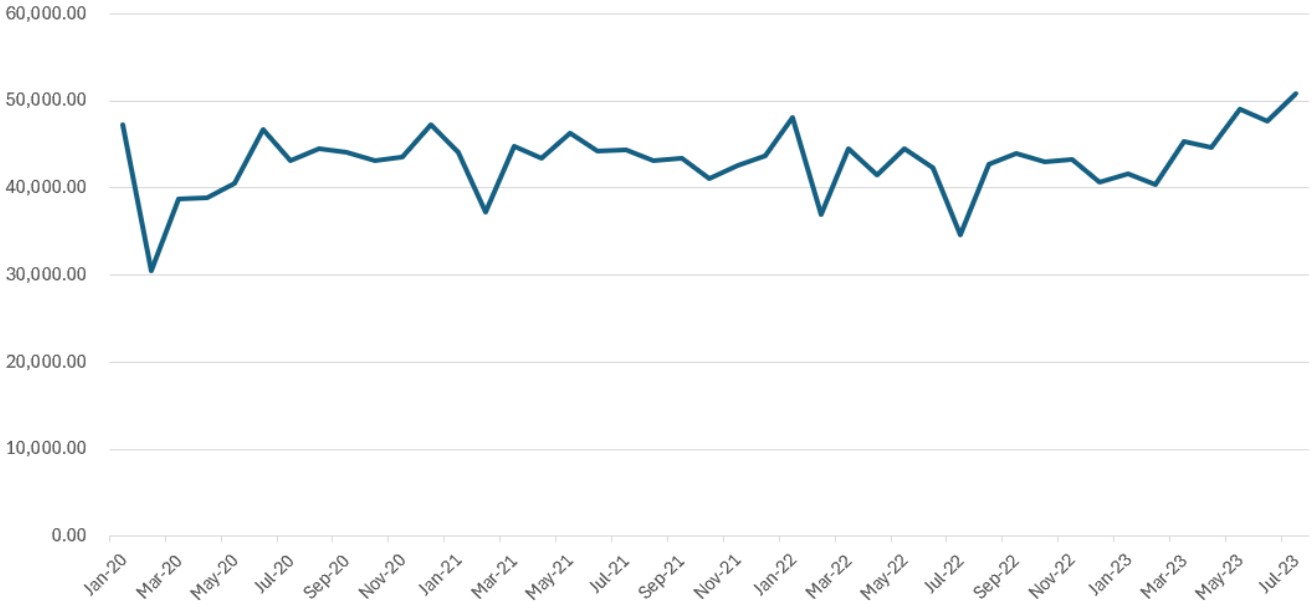

**Figure 4.** Source: Macau Environmental Protection Bureau/Received heavy metals.

By comprehensively considering these theories and considerations, better management of landfill impermeable systems can be achieved, ensuring effective handling of construction waste while minimizing environmental impact.

The construction and management of anti-seepage systems encompass various domains, including engineering construction, environmental monitoring, and quality control, among others. Effective quality management measures in these domains ensure the smooth progress of projects and the attainment of expected goals. Therefore, this paper aims to deeply investigate the quality management of the Macau landfill's anti-seepage system project, exploring how project management theories and methods can be applied in real projects to enhance project quality, reduce risks, and offer valuable lessons for similar endeavors.

### 1.3. Synthesis of Domestic and International Research

Over the past few decades, the field of quality management for landfill anti-seepage system projects has garnered extensive attention and research. Scholars and experts from around the world have extensively discussed and researched quality management methods and practices for similar projects. Internationally developed countries, such as the United States, Canada, and Australia, have established relatively comprehensive quality management systems for landfill anti-seepage system projects, accumulating rich experience.

In China, with the increasing awareness of environmental protection, research on landfill anti-seepage system projects has been on the rise. Different regions have proposed various solutions based on distinct geological, climatic, and environmental characteristics. For instance, the methods used in the anti-seepage system projects in Guangdong Province to cope with extreme weather conditions, such as typhoons and heavy rainfall, are noteworthy [3]. Simultaneously, some studies focus on the application of novel materials, such as geosynthetic liners and biodegradable materials, to enhance the effectiveness and sustainability of anti-seepage systems [4].

In the study of anti-seepage system projects, both internationally and domestically, various quality management methods and experiences have been employed to ensure their successful execution. International insights highlight the significance of early-stage geological surveys and environmental assessments in upholding project quality. Additionally, a scientifically sound design and meticulous material selection are pivotal for the functionality of anti-seepage systems [3,4].

During project implementation, a well-structured construction plan and efficient management processes play a vital role in coordinating resources, guaranteeing construction quality, and maintaining progress. Some domestic projects have embraced modern engineering technologies, such as remote sensing and Geographic Information Systems (GIS), for real-time monitoring and project management [5].

Furthermore, quality inspections and assessments are indispensable. The utilization of non-destructive testing methods, such as electrical resistivity testing and ground-penetrating radar, allows for continuous monitoring, enabling timely issue identification and resolution. Regular quality assessments contribute to a repository of lessons learned, offering guidance for future projects.

### 1.4. Basic Theory of Project Management

1.4.1. Explaining the Basic Concepts and Principles of Project Management

Project management, as an interdisciplinary management approach, aims to plan, organize, control, and supervise project implementation to achieve project objectives. In project management, some fundamental concepts and principles are essential to ensure the quality, schedule, and budget of the project [5].

Clarity of project objectives: The primary task of project management is to define project objectives. This includes specifying the project's scope, goals, deliverables, etc., ensuring that all stakeholders have a clear understanding of the expected outcomes [5].

Planning and control: Project management relies on detailed planning, including time schedules, resource plans, cost plans, and more. Planning not only guides the project but also provides the foundation for monitoring and controlling project progress [6].

Risk management: Project management emphasizes the identification, assessment, and response to risks. During project implementation, various uncertainties may arise [6]. Effective risk management strategies can mitigate the impact of risks on project quality and schedule.

Team collaboration: Project management places emphasis on teamwork, encouraging collaboration among experts from different domains. Effective communication and collaboration help reduce misunderstandings and enhance project efficiency.

### 1.4.2. Introducing Project Management Theories Relevant to Landfill Anti-Seepage System Projects

In landfill anti-seepage system projects, appropriate project management theories can help optimize resource utilization, improve project quality, and reduce risks. The following are some project management theories that may be applicable to such projects:

Agile project management: Agile methods emphasize flexibility and rapid response to changes. In landfill anti-seepage system projects, environmental factors and engineering conditions may change. Adopting agile methods can better address these variations [7].

PRINCE2 (Projects IN Controlled Environments): PRINCE2 is a structured project management method suitable for complex projects. It emphasizes phased management, clear roles, and responsibilities, which are especially crucial in anti-seepage system projects [8].

Risk management theory: Due to the complex issues involved in landfill anti-seepage system projects, such as environmental protection and resource utilization, risk management theory can help identify and address potential risks [9].

When introducing these project management theories, adjustments and applications need to be tailored according to the specific characteristics of the project.

## 2. Overview of the Landfill Containment System Project

*2.1. Describing the Scale, Objectives, and Key Components of the Macau Landfill Anti-Seepage System Project*

As an urban special administrative region with limited land resources, waste management poses significant challenges in the Macau region. The Macau landfill anti-seepage system project aims to ensure environmental safety and sustainable development during waste burial processes. The scale and objectives of this project hold crucial significance for environmental protection and sustainable development in Macau.

The project's scale is determined by the amount of waste generated and landfill requirements in the Macau region. The primary objectives of the project include [10] (Figure 5):

Environmental protection: Ensuring that buried waste does not contaminate soil and groundwater, maintaining ecological balance and the health of ecosystems.

Sustainable development: Providing sustainable environmental and resource solutions for future generations through effective waste management.

Key components of this project include:

Seepage layer design: Designing appropriate seepage layers to prevent harmful substances from waste leaking into soil and groundwater.

Isolation layer design: Ensuring that post-burial waste does not adversely impact the surrounding environment.

Drainage system: Designing a drainage system to manage rainwater and leachate, reducing environmental risks.

Environmental monitoring system: Establishing a real-time monitoring system to track environmental changes during waste burial.

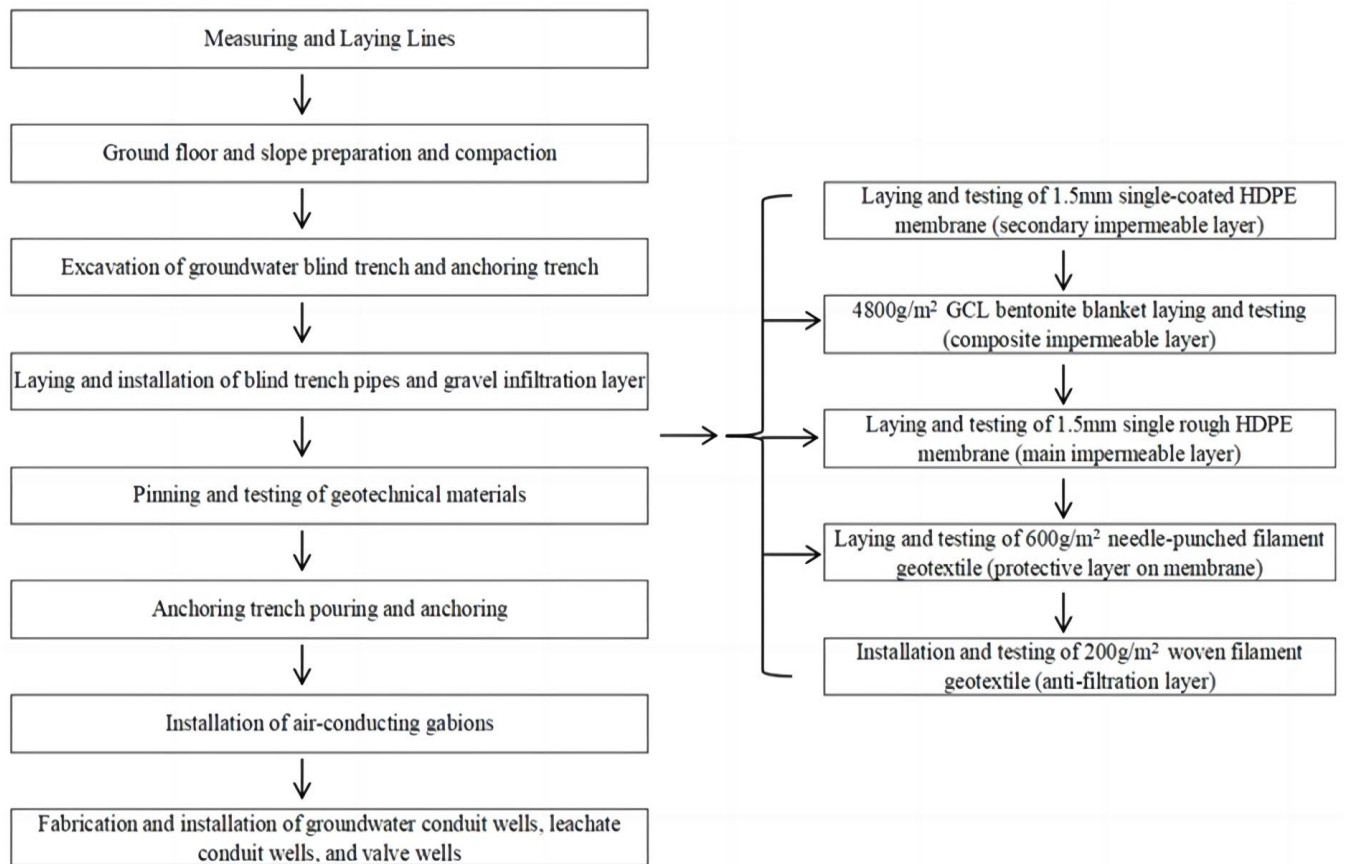

**Figure 5.** Construction process for impermeable systems.

### 2.2. Emphasizing the Project's Complexity and Risk Factors

The Macau landfill anti-seepage system project faces challenges characterized by complexity and diversity. Firstly, the unique geographical environment of Macau, characterized by a humid climate and frequent typhoons, adds to the difficulty of project implementation. Secondly, significant fluctuations in groundwater levels and complex geological conditions require thorough consideration in the design. Additionally, the diversity in waste types and characteristics necessitates the use of appropriate materials and technologies to address varying scenarios.

Project complexity also involves multidisciplinary coordination and collaboration, requiring expertise from engineers, geologists, environmental specialists, and more. Moreover, the project may be influenced by policies, social concerns, and media scrutiny, necessitating communication and collaboration with various stakeholders.

Risk factors cannot be overlooked. Given the project's involvement in environmental protection and public interests, any issues that arise could lead to significant environmental and societal impacts. Therefore, risk identification, assessment, and response strategies are crucial in project management and implementation [11].

### 3. Study of Construction Impact Factors

### 3.1. Exploring Factors Affecting the Construction Quality of Landfill Anti-Seepage Systems, Such as Geological Conditions and Material Selection

The construction quality of landfill anti-seepage system projects is influenced by various factors, among which geological conditions and material selection are important contributors to construction quality.

Geological conditions: The complexity of geological conditions significantly impacts the design and construction of anti-seepage systems. Factors such as fluctuations in groundwater levels, soil types, and layers influence the arrangement of seepage layers

and the design of drainage systems. Therefore, detailed geological surveys and analyses must be conducted during the project's initial phases to ensure scientific and feasible designs [12–14].

Material selection: The choice of materials for seepage and isolation layers directly affects project quality and effectiveness. Appropriate materials ensure the impermeability of seepage layers and the stability of isolation layers. Material selection also needs to consider aspects such as environmental friendliness, sustainability, and cost-effectiveness to balance long-term benefits and costs of the project [15–17].

### 3.2. Analyzing the Impact of These Factors on Project Progress and Quality

The extent of impact from these factors varies based on project characteristics. Different geological conditions can lead to varying design and construction difficulties, necessitating different techniques and methods. Improper material selection may lead to underperformance of the anti-seepage system and potential environmental issues.

The impact of geological conditions and material selection on project progress is also evident. Complex geological conditions might require more time for surveys and analysis, affecting project timelines. Material selection may involve procurement and supply challenges, potentially causing construction delays if appropriate materials are not obtained in a timely manner.

During project implementation, changes in geological conditions and material selection could lead to project adjustments and modifications, further emphasizing the complexity of construction impact factors. Thus, project teams need the ability to adapt to changes swiftly to ensure the project's smooth progression and quality assurance [18].

## 4. Quality Control Plan

### 4.1. Proposing a Comprehensive Quality Control Plan, including Quality Objectives, Standards, and Processes

In the landfill anti-seepage system project, a quality control plan is a crucial element to ensure that the project is completed according to expected standards (Figure 6). Here is an overview of a quality control plan for reference:

- Quality objectives

The primary objective of the landfill anti-seepage system project is to ensure that the construction quality aligns with both national and local regulations. By doing so, this will effectively protect the environment and ecosystems surrounding the landfill [19]. To achieve this objective, several specific goals will be pursued. These include ensuring that the seepage layer thickness meets the required criteria, guaranteeing the physical and chemical properties of materials used in the construction are appropriate, designing and implementing an effective drainage system, and evaluating the efficiency of the drainage system in preventing seepage;

- Quality standards

To maintain and achieve the desired construction quality, a set of quality standards will be established for the landfill anti-seepage system project. These standards will outline the necessary criteria for the seepage layer thickness. Additionally, they will specify the physical and chemical properties that materials must meet to be considered suitable for use in the construction process [20]. The standards will also dictate the design requirements for the drainage system, such as the placement and spacing of drainage elements, as well as the necessary slope or gradient for efficient water flow. Furthermore, criteria will be set to assess the effectiveness of the drainage system in preventing seepage;

- Quality processes

The quality processes for the landfill anti-seepage system project will consist of several stages, each with its own set of activities and objectives;

- Initial preparation

During this stage, comprehensive geological surveys and environmental assessments will be conducted. This process aims to ensure that the project designs take into account all relevant geological and environmental factors that may affect the construction quality. The surveys will identify potential risks, such as unstable soil conditions or sensitive ecological habitats, enabling appropriate measures to be incorporated into the project plan [21];

- Design phase

The design phase involves developing detailed engineering designs that specify the technical indicators and quality requirements. These designs will outline the specific construction methods to be followed and provide technical guidelines for the materials and equipment to be used [9]. They will also consider factors such as load-bearing capacity, structural stability, and long-term durability;

- Pre-construction phase

During the pre-construction phase, construction plans will be prepared, taking into account the design specifications and technical requirements. This phase also involves procuring the necessary materials, equipment, and personnel to carry out the construction activities effectively. Proper planning and resource allocation are essential to ensure that all required resources are available when needed [22,23];

- Construction process

The construction process will be closely monitored to ensure that every step adheres to the design and quality requirements. Real-time monitoring will enable early detection of any deviations or issues, allowing for prompt corrective actions to be taken. Documentation of the construction process will be maintained for future reference and quality assurance purposes, ensuring that the project is implemented as planned [24,25];

- Quality inspections

Regular quality inspections will be conducted throughout the construction process. These inspections will include material tests to verify their suitability and compliance with quality standards. Additionally, construction quality spot checks will be carried out to verify that the actual construction aligns with the design and meets the required quality criteria. This will involve checking dimensions, structural integrity, and proper installation of drainage elements, among other aspects. Any discrepancies or non-compliance will be addressed promptly, ensuring that the desired construction quality is achieved [26,27];

- Environmental monitoring

To monitor the impact of the waste burial on the environment, an environmental monitoring system will be implemented. This system will track and assess any changes in environmental conditions during the waste burial process. This will allow for the timely identification and addressing of any issues that may arise. Monitoring parameters may include air quality, water quality, soil conditions, and potential contamination risks. By closely monitoring the environment, the project can ensure that the landfill anti-seepage system is effectively protecting the surroundings and ecosystems [28,29];

- Quality assessment

Upon completion of the project, a comprehensive quality assessment will be conducted. This assessment will evaluate the overall quality of the construction process and outcomes. Lessons learned from the project will be summarized, highlighting both successful practices and areas for improvement. The assessment will also provide guidance for similar future projects, ensuring continuous improvement in construction quality and environmental protection in the field of waste management [27].

| Quality Planning | → | Quality Control | → | Quality Inspection | → | Quality Improvement |
|---|---|---|---|---|---|---|

**Figure 6.** Quality Management Process.

*4.2. Emphasizing the Importance of Implementing Quality Control in All Project Phases*

In the landfill anti-seepage system project, quality control should extend beyond the construction phase to encompass all stages of the project. From initial preparation to design, construction, and environmental monitoring, rigorous quality control is essential.

During the initial preparation phase, conducting geological surveys and environmental assessments is crucial. Quality control plays a significant role in ensuring the accuracy and completeness of these data. This guarantees reliable bases for subsequent design and construction. By diligently reviewing and verifying the gathered information, potential errors and inaccuracies can be eliminated, thus minimizing risks during later stages.

In the design phase, quality control becomes vital in ensuring the scientific and rational nature of the design plans. It involves a thorough examination of each detail, ensuring that technical indicators and quality standards are explicitly specified. Through meticulous supervision, any potential design flaws or oversights can be identified and rectified, enhancing the overall functionality and efficiency of the anti-seepage system.

During the construction process, quality control becomes even more crucial. It involves meticulous material selection and acceptance, guaranteeing that only high-quality materials are used in the project. Not only does this contribute to the overall durability and effectiveness of the system, but it also ensures that all construction activities comply with established standards. Having a robust construction quality management system in place helps identify and rectify any deviations from the approved designs or standards, thus ensuring that the project is completed to the highest quality and safety standards.

The implementation of an environmental monitoring system is another key aspect of quality control. With such a system in place, real-time tracking of environmental changes becomes possible. This allows project managers to proactively address any emerging environmental issues, ensuring minimal impact on the surrounding ecosystem. Regular monitoring and analysis enable prompt identification of any changes in water quality, potential leaks, or adverse impacts on nearby habitats. By promptly addressing such issues, the project can minimize environmental damage and demonstrate its commitment to sustainability.

Emphasizing the importance of implementing quality control in all project phases is essential. Doing so helps prevent issues from magnifying or compounding in later stages. By addressing potential concerns upfront, the project team can identify and mitigate risks before they escalate, thus ensuring that the project achieves its expected quality objectives. Continuous monitoring and steadfast adherence to established quality control measures guarantee the longevity, effectiveness, and safety of the landfill anti-seepage system.

## 5. Discussion

*5.1. Project Construction Quality and Control*

5.1.1. Exploring Quality Management Strategies and Practices during Actual Construction

In addition to the mentioned quality management strategies and practices, there are several more that can be implemented in the practical construction process of the landfill anti-seepage system project. These include:

1.  Quality assurance plan: Develop a comprehensive quality assurance plan that outlines all the quality measures, procedures, and responsibilities. This plan should also include a systematic documentation process to ensure all quality-related activities are recorded;
2.  Training and certification: Provide necessary training and certification programs to the construction team members. These programs should focus on enhancing their

skills and knowledge related to quality management and construction techniques, ensuring that they are equipped to handle the project requirements effectively;

3.  Risk assessment and mitigation: Conduct a thorough risk assessment to identify potential risks and hazards that may impact the construction quality. Develop appropriate risk mitigation strategies and implement them throughout the construction process;

4.  Supplier qualification: Implement a supplier qualification process to ensure that all the materials and equipment used in the construction meet the required standards. This process should involve evaluating suppliers based on their track record, quality control measures, and adherence to industry standards;

5.  Non-conformance management: Implement a non-conformance management process to address any deviations or non-compliance during the construction process. This process should include formal procedures for documenting, investigating, and resolving non-conformance issues;

6.  Continual improvement: Establish a culture of continual improvement within the project team by encouraging feedback and suggestions from all stakeholders. Regularly review the construction process and identify areas for improvement to enhance the overall quality of the project;

7.  Post-construction evaluation: Conduct a post-construction evaluation to assess the effectiveness of the quality management strategies and practices implemented. This evaluation should involve reviewing the construction documentation, conducting site inspections, and gathering feedback from stakeholders to identify any lessons learned and areas for further improvement.

By implementing these additional quality management strategies and practices, the landfill anti-seepage system project will be able to ensure a higher level of construction quality, thereby enhancing its overall effectiveness and durability.

5.1.2. Emphasizing the Importance of Monitoring, Evaluating, and Adjusting Quality Control Measures

Quality control during construction requires continuous monitoring, evaluation, and adjustment. Monitor the implementation of quality control measures to ensure their effectiveness and appropriateness. Evaluate the quality status during the construction process, compare actual conditions with expected objectives, and promptly identify any deviations [30–35].

If issues arise during actual construction, quality control measures need adjustments. While addressing problems, consider a holistic approach encompassing engineering techniques, material performance, environmental conditions, etc., and implement appropriate corrective and improvement measures. Additionally, maintain thorough documentation and reporting for subsequent analysis and summarization.

Emphasizing the importance of monitoring, evaluating, and adjusting quality control measures aids in continuously optimizing project quality during the construction process, ensuring the project is completed in line with expected standards (Figures 7 and 8).

In the forthcoming sections, we will delve into quality inspections of the anti-seepage system project, discussing how to effectively monitor and evaluate project quality (Table 3).

*5.2. Quality Control and Assessment*

5.2.1. Introducing Quality Inspection Methods and Tools, including Non-Destructive Testing and Sampling

In the context of the landfill anti-seepage system project, quality inspection plays a pivotal role in ensuring that construction quality meets the anticipated standards. The following methods and tools are commonly employed for quality inspection:

Non-destructive testing (NDT): NDT techniques are utilized to gather information about material properties and structural integrity without causing any damage to the engineering structure. In the anti-seepage system project, innovative methods such as electrical resistivity testing and ground-penetrating radar can be employed to monitor

the quality of the anti-seepage layer in real time. These advanced techniques enable the project team to detect any potential weaknesses or defects in the system, ensuring its long-term effectiveness.

Sampling inspection: In order to evaluate the construction quality, sampling inspection is carried out by testing samples taken from the construction site. These samples effectively reflect the overall construction quality to a certain extent. Well-designed sampling inspection methods not only improve the efficiency of the quality inspection process but also reduce inspection costs. By carefully selecting and testing samples, it becomes possible to identify any deviations from the anticipated standards and take corrective measures promptly.

**Table 3.** Quality control points.

| No. | Control Items | | Main Control Points | Responsible Person for Control | Control Basis | Remarks |
|---|---|---|---|---|---|---|
| 1 | Groundwater Diversion Project | Depot Site | Levelling longitudinal and transverse slope, compaction, bearing capacity | Surveyor, tester | Construction drawings, work Instructions, technical delivery documents, construction specification | |
| | | Blind Ditch | Size, spacing, burial depth | Surveyor, quality inspector | | |
| | | Gravel deflector layer | Gravel particle size, laying method, Laying thickness, anti-filtration layer thickness | Constructer, quality inspector | | |
| | | Drainage wells | Groundwater control, axis and elevation on piles, cement collapse, pouring, maintenance time | Project general engineer, quality inspector, constructor | | |
| 2 | Horizontal seepage control system project | HDPE membrane | Thickness, verticality of cut edges, laying sequence joint direction location quantity: welding method time between welding machine songs. lap width ambient temperature, leakage detection of welds, weld qualification rate, finished product protection | Constructor, quality inspector | Work instructions, technical delivery documents, construction specifications | |
| | | Geotextile | Connection method, lap width, appearance | Constructer, quality inspector | | |
| | | GCL bentonite Blanket | Paint penetration coefficient method of joining, width of joining, disposition on of joining blanket, sealing material, sealing method, moisture-proof protection measures. protection of finished products | Constructer, quality inspector | | |
| | | Composite drainage net | Surface dryness. surface cleanliness. laving method lap joint tensile and compressive strength | Constructer, quality inspector | | |
| | | HDPE pip | Cutting length welding method. adjacent pipe center line control, construction temperature, pipe opening plugging measures, welding ring height. welding cooling time | Constructer, quality inspector | | |
| | | Anchoring trench | Collapse degree. pouring, maintenance, location of expansion joints | Constructer, quality inspector | | |
| 3 | Leachate Drainage Engineering | Arti-filtration laver | Gravel particle size, laying method, key thickness | Constructer, quality inspector | Construction drawings. work instructions, technical delivery documents, construction | ______ |
| | | Drainage shaft | Axis and elevation piles, pipe welding. seepage and anti-corrosion performance | Constructer, quality inspector | | |
| 4 | Electrical Equipment And Ancillary works | Air-conducting gabions | Opening diameter. reinforcement grid diameter. grid gap Construction drawings | Constructer, quality inspector | technical briefing documents construction specifications | |

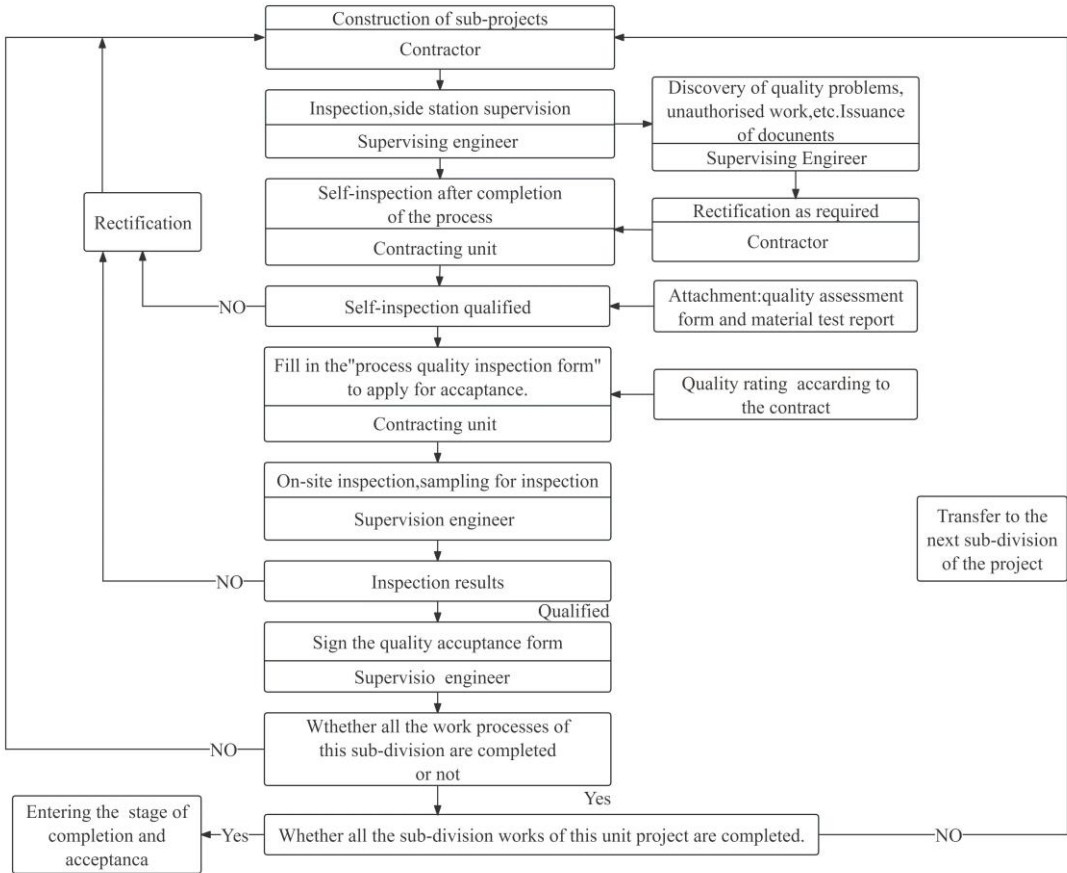

**Figure 7.** Quality control of sub-projects.

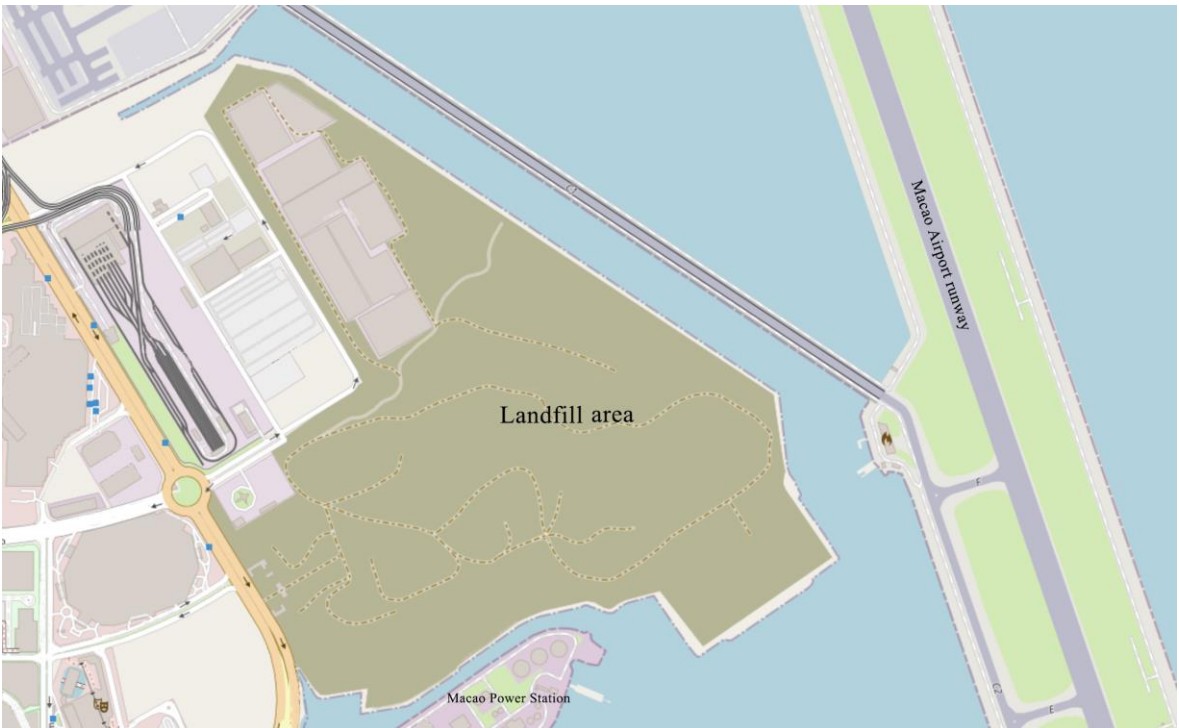

**Figure 8.** Source: The Land and Urban Construction Bureau of Macao.

On-site inspection: Regular on-site inspections are conducted to meticulously audit various aspects of the project. Expert inspection personnel thoroughly examine the materi-

als, processes, and equipment during the construction phase. By closely monitoring the construction activities, any potential issues or non-compliance with the desired standards can be identified promptly. This allows the project team to take immediate action, rectify the problems, and ensure that the construction quality is maintained at the highest level. Regular on-site inspections also provide an opportunity for the inspector to communicate with the construction team, address any concerns, and provide guidance if necessary [36–39].

In summary, quality inspection plays a vital role in the landfill anti-seepage system project. Through the utilization of advanced non-destructive testing techniques, well-designed sampling inspection methods, and regular on-site inspections, the construction quality can be monitored effectively, ensuring that it meets the expected standards (Table 4).

**Table 4.** Quality inspection task sheet.

| No. | Name of Sub-Works | Work Process | Frequency | | Remarks |
|---|---|---|---|---|---|
| 1 | Groundwater conduction and drainage system | Foundation of reservoir area | Real Time | | ______ |
| | | Blind trench | Real Time | Monthly | ______ |
| | | Groundwater conduit | Real Time | | ______ |
| | | Groundwater collection well | Real Time | | ______ |
| 2 | Horizontal impermeable system | Secondary impermeable layer | Real Time | | ______ |
| | | Composite impermeable layer | Real Time | Monthly | ______ |
| | | Primary impermeable layer | Real Time | | ______ |
| | | Protective layer on membrane | Real Time | | ______ |
| | | Anchoring trench | Real Time | | ______ |
| 3 | Leachate collection and drainage well | Leachate conduit | Real Time | Monthly | ______ |
| | | Leachate conduit well | Real Time | | ______ |
| | | Infiltration layer | Real Time | | ______ |
| | | Anti-filtration layer | Real Time | | ______ |
| 4 | Electrical equipment and ancillary works | Control box | Real Time | Monthly | ______ |
| | | Power cables | Real Time | | ______ |
| | | Air-conducting gabions | Real Time | | ______ |

### 5.2.2. Discussing How to Evaluate Project Quality and Progress to Ensure Meeting Expected Objectives

Quality assessment is a crucial component in ensuring the overall success of a project. Alongside monitoring the progress of the project, assessing its quality helps to identify areas of improvement and maintain high standards. There are several methods and indicators that can be used to assess project quality.

Firstly, setting appropriate quality indicators based on the project design and quality standards is essential. These indicators serve as benchmarks to evaluate the project's quality status. By comparing the actual construction conditions with the requirements set by the indicators, project managers can assess whether the quality objectives are being met.

Additionally, progress indicators are crucial in evaluating the project's progress. Developing a project schedule with milestones and phased objectives allows for the monitoring of planned versus actual completion against the established timeline. This helps to identify any discrepancies and take corrective actions accordingly.

The impact of the project on the environment can also be assessed through environmental monitoring data. These data include factors such as groundwater quality and soil contamination. By comparing the monitoring data with environmental standards, project managers can evaluate the environmental impact of the project and take necessary measures to mitigate any potential harm.

Regular inspections and assessments are instrumental in maintaining project quality. By conducting inspections across various aspects of the project, any issues or deviations can be identified promptly. Once identified, corrective measures should be taken promptly to prevent the escalation of these issues, thus ensuring the project stays on track and meets the desired quality standards.

Furthermore, after project completion, summarizing the project's quality and progress is crucial. This summary allows for the extraction of lessons learned, which can then be utilized to provide guidance and insights for future similar projects. By reflecting on the project's successes and challenges, project managers can continuously improve their practices and strive for better outcomes in the future.

Overall, through the continuous evaluation of project quality and progress, any issues can be promptly identified and addressed. This allows for necessary adjustments to be made, ensuring that the project meets its intended objectives. This commitment to evaluation and adjustment contributes to the maintenance of high-quality and efficient project outcomes.

## 6. Conclusions

### 6.1. Summarizing the Main Findings and Contributions of the Research

This paper offers the results of thorough research into quality management in landfill anti-seepage system projects implemented in Macau, describing different steps of project quality control from initial readings through operational responses. The paper focuses on Macau' formation as a special administrative region and creates particular challenges in addressing waste management issues, highlighting landfill administration systems and anti-seepage treatments that are effective measures of environmental protection, gaining support for sustainable development. It covers a comprehensive review of both domestic and international research on landfill anti-seepage systems, therefore, deepening the understanding about varied quality management approaches and thus making it invaluable material for the effective implementation of such projects. The article is expedient as a theoretical basis for the managed implementation of landfill anti-seepage system projects. It substantiates important project management rules and theories outlining its complexity. Outlining the Macau landfill anti-seepage system in detail, it concentrates on scale of operation and objectives to be achieved, as well as leaving no room unaffected by core parts that are being analyzed and making clear high specifics and also potential risks. Highlighting factors shaping construction quality, such as soil properties or materials applied, gives important insight into risk monitoring during the construction phase. A quality control plan that encompasses all aspects of project management increases the efficacy of its implementation. These tools include quality objectives as points of reference, standards that must be met during each phase, and procedures to signal the significance of a constant practice used in all stages along these lines. The paper talks about construction quality and control of the project and details the needed techniques and standards for ensuring righteousness in the projects. It highlights the need to undertake on-going monitoring and reorientation of quality control procedures necessary in sustaining construction standards as well achieving project targets. Non-destructive testing and sampling are methods that help ensure quality inspection and assessment at the end of all project phases, concluding the paper with a holistic approach to quality control by incorporating a comprehensive overview throughout.

### 6.2. Emphasizing the Key Role of Quality Management in Landfill Anti-Seepage System Projects

In this case, the important part that quality management has in the landfill anti-seepage system becomes evident through this project. A quality management system should indeed be quite comprehensive to cover all stages of project implementation, including preparation for the task, design solutions or choice among variants after that stage has been completed successfully and the construction itself as well as the subsequent operation with monitoring. This summary angle of quality management reaches beyond technical details;

it is a multifaceted task, requiring interdisciplinary expertise and team work. In future anti-seepage landfill projects, there will be a need to mainstream research and practice on quality management, drawing from national and local levels as well as internationally recognized experiences for it enhancement. Quality management strategies and methods should be innovative in doing so at the same time; they must also always be optimized. This can only be achieved through the use of high-quality construction and strict control over workmanship, accompanied by environmental protection objectives and environmentally friendly practices. Finally, this paper exhaustively studied quality management in "landfill anti-seepage system projects", peeking extensively into issues that academic research as well professional engineering application should consider critically before implementing. Future studies will continue to investigate and compare quality management methods with different cases or consider the application of quality management in environmental protection, resource conservation, and sustainable development.

## 7. Research Outlook

### 7.1. Proposing Future Research Directions and Possibilities

Although this paper has presented an in-depth examination of quality management practices within landfill anti-seepage system projects, further exploration may yield many avenues and possibilities for advancement:

Future research could emphasize interdisciplinarity by drawing together knowledge from engineering, geology, environmental sciences, and other fields in order to address complex project requirements effectively.

Integration across disciplines has become an established theme in environmental management. A study, "Interdisciplinary Approaches to Environmental Management", released by offers profound insight and demonstrates its significance, emphasizing its benefits. Specifically, environmental management issues often involve complex factors interwoven from geology, ecology, engineering, and environmental sciences that necessitate looking outside traditional disciplines for solutions while simultaneously gathering specialized knowledge across domains to fully comprehend and address environmental problems effectively.

One excellent example of successful interdisciplinarity can be seen through the landfill anti-seepage system project in certain regions of the US. In this project, the project management team expertly combined the knowledge and skills of geologists, environmental engineers, and ecologists. Geologists were instrumental in understanding groundwater flow and geological conditions. Environmental engineers were responsible for designing an anti-seepage system, and ecologists assessed its impact on surrounding ecological environments. All three disciplines collaborated seamlessly together, leading to more thorough project planning and execution.

The geologists provided invaluable assistance when selecting suitable anti-seepage layer materials to prevent harmful substances from leeching into soil and groundwater sources, and the ecologists ensured the project would have no ill effects on the local ecological balance during its implementation. As a result, this project not only addressed groundwater seepage issues but also protected its local ecosystem and water sources for long-term success.

This case illustrates the benefits of interdisciplinarity, such as applying knowledge from different fields to effectively address complex problems. Interdisciplinary collaboration should not just be considered an option when designing landfill anti-seepage system projects but instead should become essential to guarantee project quality and achieve environmental protection goals. Thus, more consideration should be given to it during future research or practical applications.

Smart monitoring technologies: As technology progresses, smart monitoring technologies for project quality management will become ever more crucial. Research could focus on using sensors, drones, artificial intelligence, or any other means for real-time

monitoring and data analysis to maximize precision and efficiency when it comes to quality management.

Intelligent monitoring technology has quickly become an integral component of modern environmental management, as evidenced by its wide uptake. Citations such as those provided in "Smart Technologies for Environmental Monitoring and Management" by *Environmental Science & Technology* (2020) show the numerous applications and significant advantages offered by intelligent monitoring technology in environmental monitoring—not only improving data collection efficiency but also real-time analysis to increase precision and improve precision as part of project quality management or environmental monitoring processes.

As an illustration of how intelligent monitoring technology is applied in real projects, consider Canada's landfill project, where advanced sensor technology and drones were extensively employed for monitoring leachate emissions and environmental parameters in real time. These sensors capture critical data in real time while drones are employed for aerial photography and collecting information that is then transmitted centrally for analysis and interpretation in real time.

Through using intelligent monitoring technology, this landfill project has experienced numerous advantages. First and foremost, its environmental management team can quickly recognize and respond to potential issues without waiting for traditional data reports to be generated. Secondarily, real-time analysis contributes to quality management as problems can quickly be rectified in accordance with environmental standards. Finally, intelligent monitoring technology has enhanced environmental monitoring efficiency, helping increase understanding about its effect on surrounding environments.

Intelligent monitoring technology has significantly benefited project quality as well as environmental monitoring and protection in this instance, and its successful deployment demonstrates its importance when dealing with complex environmental management tasks, providing real-time data to facilitate decision support in projects such as landfill anti-seepage system installations. Therefore, smart technology integration should become part of future landfill anti-seepage system projects to guarantee meeting both quality goals as well as environmental protection goals.

Future research could emphasize sustainability considerations in project quality management: For sustainable development, future studies might place greater focus on environmental protection, resource utilization, and social benefits to ensure effective quality management practices for project delivery.

As an illustration of how sustainability considerations should be embedded into landfill projects, one project in China offers us an ideal example. Sustainability was integrated from its inception, with emphasis being placed upon resource recovery and environmental conservation measures aimed at minimizing its adverse environmental impact while seeking sustainable development goals.

One of the key sustainability measures included in this project was resource recovery to implement processes to identify and recover recyclable materials from waste streams and reduce overall landfill volume. By doing so, valuable resources were saved, thereby creating an economy with circular resources and practices.

China's landfill project also included stringent environmental protection measures to mitigate its negative environmental impact and promote ecological balance within its surrounding community. Such measures included anti-seepage systems to avoid groundwater contamination, comprehensive leachate management strategies, and green technologies for emissions control; these were just a few examples among others. By employing them effectively, this project managed to lessen its negative environmental impacts as well as promote ecological harmony throughout.

Success of this Chinese landfill project highlights the practical implications of taking into account sustainability concerns when planning anti-seepage system projects. By emphasizing resource recovery and environmental protection, this project not only met its waste disposal goals but also aligned with wider sustainability objectives. This example

underscores the necessity of including sustainability considerations when planning and executing landfill projects, not only because this protects the environment but also because it contributes to communities' and ecosystems' long-term wellbeing. Prioritizing sustainability considerations is, therefore, integral for meeting both immediate and long-term project goals.

*7.2. Exploring Methods and Strategies for Continuous Improvement of Anti-Seepage System Project Quality Management in a Changing Environment*

In an ever-changing environment, project quality management requires ongoing improvement and adaptation. Here are some potential methods and strategies.

Learning from others: Study domestic and international cases of similar projects, learning from their quality management experiences and lessons to provide insights and inspiration for one's own project.

1. Learning from others: A multifaceted approach

In the realm of anti-seepage system projects, the value of learning from the experiences and successes of others cannot be overstated. A comprehensive review of the literature and case studies illuminates the multifaceted nature of this approach, offering profound insights into how it contributes to the enhancement of project quality and environmental protection. This multifaceted approach involves the following key elements;

2. Extensive literature review: A knowledge repository

The foundation of learning from others lies in a comprehensive literature review that draws upon a wealth of academic research, industry reports, and documented case studies. These sources collectively form a knowledge repository that encapsulates decades of anti-seepage system project experiences [35]. By meticulously analyzing this study and similar research, practitioners have at their disposal a vast pool of knowledge, which provides context, best practices, and emerging trends in anti-seepage system project management;

3. Benchmarking against success stories

Learning from others extends beyond passive knowledge acquisition; it involves benchmarking against successful projects worldwide. By evaluating projects that have achieved exemplary outcomes, project managers can distill valuable lessons and principles that transcend geographical boundaries [39–42]. The study by Loushine and Smith et al. (2006) [43] specifically underscores the significance of benchmarking against these success stories. It serves as a guidebook for understanding the underlying factors that have contributed to the triumph of anti-seepage system projects in diverse contexts. Such benchmarking exercises facilitate the identification of universal project management models and strategies that have proven effective in a variety of scenarios;

4. Nuanced perspectives: Regional and environmental variables

One of the distinctive aspects of learning from others is the opportunity to gain nuanced perspectives on the influence of regional and environmental variables. Case studies from different regions shed light on how geographical, climatic, and ecological factors can impact project quality and outcomes. These studies enable project managers to develop a heightened sensitivity to the specific challenges and opportunities presented by the project's location. For instance, a project in a coastal area may require different anti-seepage measures compared to one situated inland [44–47]. Research such as that of Smith et al. (2006) guides practitioners in navigating these nuances effectively;

5. Customized approaches: Tailoring solutions

Armed with deep-rooted insights derived from an extensive review of the literature and case studies, project managers are better equipped to tailor their approaches to fit specific project conditions. Instead of adopting one-size-fits-all strategies, they can customize solutions that address the unique challenges posed by a particular anti-seepage system project. The ability to apply knowledge gained from successful projects worldwide

to local contexts enhances the adaptability and responsiveness of project management practices. By steering clear of pitfalls and integrating sustainable techniques drawn from these lessons, practitioners can navigate the complex terrain of anti-seepage system projects with confidence.

In conclusion, learning from others through the multifaceted approach of a literature review, benchmarking against success stories, understanding regional and environmental variables, and customizing solutions is a linchpin in advancing the quality management of anti-seepage system projects. This approach capitalizes on the collective wisdom of the global community, transcending borders and disciplines to fortify project managers with the knowledge and insights needed to ensure the success of anti-seepage system initiatives.

**Funding:** This research received no external funding.

**Data Availability Statement:** The data on which the study is based were accessed from a repository and are available for downloading through the following link: https://www.dspa.gov.mo/index.aspx (accessed on 13 September 2023).

**Conflicts of Interest:** The authors declare no conflict of interest.

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
