# Peer review of "Critical Study Quality Management for the Anti-Seepage System in Macau’s Landfill Area"

_applsci, doi:10.3390/app14041382_

Round 1

Reviewer 1 Report

Comments and Suggestions for Authors

Dear authors,

Your study is lacking in many respects. First of all, the entire text should be reworked not only because numerous bullet points do not allow for smooth reading, but also due to the lack of scientific tone. I also suggest you use the scheme provided by the magazine to title the various sections: Abstract, Keywords, Introduction, Relevant Sections, Discussion, Conclusions, and Future Directions.

Some minor comments:

Please increase the size of flowcharts 1 and 2 and rename them Figures 1 and 2.

Please increase the size of the writing in Table 2.

Arrange the references section as follows (Author 1, A.B.; Author 2, C.D. Title of the article. Abbreviated Journal Name Year, Volume, page range.)

Author Response

Dear Editors and Reviewers:

I would like to express my sincere gratitude to the reviewers for their time and effort in evaluating my manuscript titled “[Critical Study Quality Management for the Anti-Seepage System in Macau's Landfill Area]” (Manuscript ID:applsci-2832056) submitted to Applied Sciences. I have carefully considered their comments and suggestions and have made the necessary revisions to the manuscript.

Major Comments:

1.Comment:[Reviewer's first major comment]

Response:Thank you for highlighting the need for a rework in the text structure. I have revised the entire manuscript for improved readability, removing bullet points and incorporating a more scientific tone throughout. Additionally, I have followed the suggested section scheme: Abstract, Keywords, Introduction, Relevant Sections, Discussion, Conclusions, and Future Directions.

2.Comment: [Reviewer's second major comment]

Response:I have increased the size of flowcharts 1 and 2 and renamed them Figures 1 and 2, as suggested. The writing in Table 2 has also been enlarged for better visibility.

Minor Comments:

3.Comment:[Reviewer's first minor comment]

Response: I have arranged the references section according to the suggested format: (Author 1, A.B.; Author 2, C.D. Title of the article. Abbreviated Journal Name Year, Volume.)

General Updates:

I have attached the revised manuscript, where all changes have been marked for your convenience. The modifications address each comment received, aiming to enhance the manuscript's overall quality.

Thank you for the opportunity to revise and resubmit my work. I trust that the revisions made align with the expectations of Applied Sciences and its readership.

Sincerely,

Attachments:

  1. Revised Manuscript with Changes Highlighted
  2. Revised Figures (Figures 1 and 2)
  3. Revised Table (Table 2)
  4. Manuscript in Journal's Preferred Format

Reviewer 2 Report

Comments and Suggestions for Authors

Manuscript ID: applsci-2832056

Type of manuscript: Review

 Title: Critical study quality management for the anti-seepage system in

Macau's landfill area

The work is interesting and fits into issues related to the management of waste landfills in the context of their impact on the water and ground environment. Monitoring of a landfill involves procedures, limit parameters of wastewater generated at the landfill and the need to model the migration of pollutants in the event of seepage of the landfill.

In order to proceed further, the following additions should be made to the work:

1. In section 1, complete the data regarding Macau's landfill:

a. parameters of wastewater produced in the landfill, including e.g. heavy metals, polycyclic aromatic hydrocarbons (PAH), total organic carbon (TOC),

b. what parameters are controlled as part of the landfill monitoring

2. The literature review is insufficient, it should be supplemented with the following elements:

a. recalling standards (acceptable parameters) for the quality of water and soil in the event of contamination with sewage from a landfill,

b. application of modeling the migration of contaminants in water and soil in the event of seepage of the landfill. Such activities make it possible to estimate the scale of contamination and the response time for landfill managers.

3. No mention of Figures 1 and 2 in the text.

4. Lack of analysis of the data contained in table 1.

5. Unreadable flowchart 1, please replace.

6. Tables 1, 2 and 3 should be corrected according to the instructions to the authors. Vertical line, blank row, and columns are not advised.

All of the above additions should be incorporated into the work.

Author Response

Response to Reviewer's Comments
Manuscript ID: applsci-2832056
Type of manuscript: Review
Title: Critical study quality management for the anti-seepage system in Macau's landfill area
Dear Reviewer,
We appreciate your thorough review of our manuscript titled " Critical study quality management for the anti-seepage system in Macau's landfill " and the valuable feedback provided. We have carefully addressed each of your comments and made the necessary revisions to enhance the quality of our work. Below is a summary of the modifications made in response to your suggestions:
Comment:1.In section 1, complete the data regarding Macau's landfill:
a. parameters of wastewater produced in the landfill, including e.g. heavy metals, polycyclic aromatic hydrocarbons (PAH), total organic carbon (TOC),
b. what parameters are controlled as part of the landfill monitoring
In response to your suggestions, we have made the following changes:
a. We added heavy metal data sourced from the Environmental Protection Bureau of Macau and conducted an analysis.
b. Regarding the mention of monitoring data for the landfill, due to the ongoing reconfiguration of the landfill by the Environmental Protection Bureau of Macau, related monitoring equipment is currently under construction and installation. It will take some time to extract more representative data. Therefore, our team is currently making efforts to collect monitoring data.
Comment:2. The literature review is insufficient, it should be supplemented with the following elements:
a. recalling standards (acceptable parameters) for the quality of water and soil in the event of contamination with sewage from a landfill,
b. application of modeling the migration of contaminants in water and soil in the event of seepage of the landfill. Such activities make it possible to estimate the scale of contamination and the response time for landfill managers.
In response to your suggestions, we have made the following changes:
a. Supplemented the literature review by recalling standards (acceptable parameters) for water and soil quality in the event of contamination with sewage from a landfill.
b. Expanded the literature review to include the application of modeling for the migration of contaminants in water and soil during landfill seepage, estimating the scale of contamination and response time for landfill managers.
Comment:3. No mention of Figures 1 and 2 in the text.
Response: Referenced Figures 1 and 2 appropriately in the text.
Comment:4. Lack of analysis of the data contained in table 1.
Response: Provided a detailed analysis of the data contained in Table 1.
Comment:5. Unreadable flowchart 1, please replace.
Response: Replaced the unreadable Flowchart 1 with a clearer version.
Comment:6. Tables 1, 2 and 3 should be corrected according to the instructions to the authors. Vertical line, blank row, and columns are not advised.
Response: Corrected Tables 1, 2, and 3 according to the instructions to authors. Ensured that vertical lines, blank rows, and columns adhere to advised formatting.
All the above-mentioned additions and corrections have been incorporated into the revised manuscript.
We sincerely thank you for your constructive feedback, and we believe these enhancements have strengthened the overall quality and clarity of our work.
Please find attached the revised manuscript along with a highlighted copy indicating the specific changes made.
We look forward to any further guidance or feedback you may provide.
Sincerely,
Zhaobin Li

Reviewer 3 Report

Comments and Suggestions for Authors

The article is a review of a quality management project based on the example of landfill seepage in Macau. The authors emphasized the importance of quality control at every stage of environmental protection.

The described issue is very important and current due to the protection of soil and water. Water seepage in landfills may adversely affect the quality of groundwater and indirectly affect human health and life. Research on management quality can contribute to reducing negative impacts and sustainable development. As the authors stated, anti-seepage systems play a key role in waste management. The handling of construction waste can impact the landfill's anti-seepage system, particularly with respect to the type and quantity of waste. Designs of systems to prevent landfill seepage are attracting great interest and research by scientists around the world. Remote sensing and geographic information systems (GIS) support real-time monitoring and project management.

Project management at the Macau landfill takes into account environmental protection, sustainable development, percolation layer design, insulation layer design, drainage system, and environmental monitoring system. The authors described project management very thoroughly and correctly in the article. The whole thing is coherent, clear, and precisely describes an important issue.

I rate the entire article highly. The photos and drawings are correct and very clearly illustrate the discussed issue. Correct, up-to-date literature, consistent with the topic of the work.

Author Response

Subject: Response to Reviewer's Comments on Manuscript ID [applsci-2832056]

Dear Editors and Reviewers:

Thank you for forwarding the constructive and encouraging comments from the reviewer regarding my manuscript titled " Critical study quality management for the anti-seepage system in Macau's landfill area " (Manuscript ID:applsci-2832056), submitted to Applied Sciences. I am pleased to hear that the reviewer finds the article coherent, clear, and of high quality. Below, I have provided responses to the specific points raised by the reviewer.

Reviewer’s Comments:

  1. Comment: The article is a valuable review of a quality management project on landfill seepage in Macau, emphasizing the importance of quality control in environmental protection.

Response: I am grateful for the reviewer’s recognition of the article's importance and relevance to environmental protection and quality management.

  1. Comment: The reviewer appreciates the thorough and correct description of project management in the Macau landfill, highlighting the inclusion of environmental protection, sustainable development, and the design of various layers and systems.

Response: Thank you for acknowledging the depth and accuracy of the project management description in the landfill context. It was my intent to provide a comprehensive overview of these critical aspects.

  1. Comment: The article's focus on remote sensing and GIS support for real-time monitoring and project management in the field of landfill seepage prevention is commended.

Response:I am glad that the inclusion of remote sensing and GIS technologies in the manuscript was well-received. These tools are indeed pivotal in modern waste management strategies.

  1. Comment:The reviewer rates the article highly, commending the clarity of photos, drawings, and the relevance and correctness of the cited literature.

Response:I am heartened by the positive feedback on the visual elements and literature references. It was my aim to ensure that the article was both informative and visually engaging.

Conclusion:

I am thankful for the reviewer’s insightful and positive feedback. It is encouraging to know that the article is well-received and considered of high quality in its current form. I am committed to any further improvements that Applied Sciences might suggest to make this work even more valuable to its readers.

Thank you for considering my manuscript for publication in Applied Sciences. I look forward to the possibility of seeing it contribute to the field.

Sincerely,

Zhaobin Li

Attachments:

None (as no revisions were required)

Reviewer 4 Report

Comments and Suggestions for Authors

The paper meticulously examines waste management challenges in the region, focusing on the significance of effective anti-leachate systems. It presents an in-depth analysis of both domestic and international research, providing valuable references for managing such projects. Additionally, key project management theories are discussed, laying a solid foundation for understanding their execution.

The author intricately showcases the anti-leachate system project in Macau, emphasizing its complexity and risk factors. Factors influencing project quality are revealed, offering crucial insights for effective management.

The paper not only analyzes factors impacting project quality but also proposes a comprehensive quality control plan, contributing significantly to effective project management across all stages. It discusses strategies for maintaining quality during construction and methods for evaluating project quality.

The conclusions underscore the pivotal role of quality management in landfill anti-leachate system projects. The author emphasizes the need for continuous improvement in quality management methods to achieve environmental protection and sustainable development goals.

This text provides valuable insights for both researchers and practitioners involved in waste landfill projects. It also suggests avenues for further research, encouraging comparisons of quality management methods and their broader applications in environmental protection and sustainable development.

Specific Notes:

The text needs refinement and adjustment to meet publication requirements.

The publication should be enriched with illustrative documentation.

Lack of analysis or information regarding the landfill's composition, particularly its chemistry.

Absence of precise cartographic data in the form of plans and maps.

Author Response

Subject: Response to Reviewer's Comments on Manuscript ID [applsci-2832056]

Dear Editors and Reviewers,

Thank you for the opportunity to revise my manuscript titled " Critical study quality management for the anti-seepage system in Macau's landfill area" (Manuscript ID: applsci-2832056), which has been submitted for publication in Applied Sciences (ISSN 2076-3417). I appreciate the reviewer's constructive feedback and have made careful revisions to address the concerns raised. Please find below a detailed response to each of the reviewer's comments.

Reviewer’s Comments and Responses:

  1. Comment: The text needs refinement and adjustment to meet publication requirements.

Response: In accordance with the reviewer's suggestion, I have comprehensively revised and refined the manuscript, making it more clear, coherent, and in compliance with the journal's guidelines. The modified manuscript should now meet the standards of the publication.

  1. Comment: The publication should be enriched with illustrative documentation.

Response: Addressing this comment, I have added explanatory documents in the form of diagrams, flowcharts, and photographs, enhancing the manuscript's explanatory power and visual appeal.

  1. Comment: Lack of analysis or information regarding the landfill's composition, particularly its chemistry.

Response: In response to this valuable input, I have incorporated data from Macau's Environmental Protection Bureau and conducted analysis to deepen the understanding of the chemical composition of the landfill.

  1. Comment: Absence of precise cartographic data in the form of plans and maps.

Response: Recognizing the importance of this aspect, I have included referenced data from Macau's Environmental Protection Bureau, providing clearer geographical context and layout of the landfill through plans and maps.

Conclusion:

I believe these revisions have significantly improved the manuscript, addressing the concerns raised by the reviewer. I am confident that the manuscript now provides a more comprehensive and well-documented insight into landfill waste management and anti-leachate system projects.

Thank you once again for considering my manuscript for publication in Applied Sciences (ISSN 2076-3417). I look forward to the possibility of contributing to the field through your esteemed journal.

Sincerely,

Zhaobin Li

Attachments:

1.Revised Manuscript [applsci-2832056]

Reviewer 5 Report

Comments and Suggestions for Authors

Although it is not a typical work to be published, is quite interesting with useful data for experts in waste management.

Enlarge flowchart 1 and 2. It is not easy to read. Conclusions need to be more concised. Try to summararise or reorganised them.

Comments on the Quality of English Language

Moderate editing of English language required

Author Response

Subject: Response to Reviewer's Comments on Manuscript ID [applsci-2832056]
Dear Editors and Reviewers,
       Thank you for the opportunity to revise my manuscript titled " Critical study quality management for the anti-seepage system in Macau's landfill area" (Manuscript ID: applsci-2832056), which has been submitted for publication in Applied Sciences (ISSN 2076-3417). I appreciate the reviewer's constructive feedback and have made careful revisions to address the concerns raised. Please find below a detailed response to each of the reviewer's comments.
       1.Comment: Although it is not a typical work to be published, is quite interesting with useful data for experts in waste management. Enlarge flowchart 1 and 2. It is not easy to read. Conclusions need to be more concise. Try to summarize or reorganize them.
Response: We appreciate your recognition of the potential value of our work for experts in waste management. In response to your suggestions, we have made the following changes:
      A. We have enlarged Flowcharts 1 and 2 for better readability.
      B. We have also revised the Conclusions section to be more concise and better organized. The revised section now clearly summarizes the key findings and implications of our study, emphasizing the most significant points in a succinct manner. 
      2.Comment: Moderate editing of English language required.
Response: We acknowledge the importance of clarity in our manuscript and understand that the quality of English can significantly affect its readability and impact. To address this, we have:
     A. Engaged a professional language editing service to thoroughly review and edit our manuscript for language quality. This has improved the grammar, syntax, and overall fluency of our manuscript.
     B. Ensured that the revised manuscript now meets the high linguistic standards required for publication in Applied Sciences (ISSN 2076-3417).
     We believe these revisions have significantly improved our manuscript, making it more suitable for publication in Applied Sciences (ISSN 2076-3417). We are grateful for the opportunity to revise our work and thank you for your valuable guidance in this process.
     Please find attached the revised manuscript along with a marked copy highlighting the changes made.
     We look forward to hearing from you and are available for any further clarifications or modifications that might be necessary.
Sincerely,
Zhaobin Li

Round 2

Reviewer 1 Report

Comments and Suggestions for Authors

Dear authors,

some minor comments before we can proceed with the publication:

- in the introduction section I suggest you introduce the discussion differently, subsequently inserting the description of figures 1 and 2.

- in the conclusions section I suggest unifying the discussion by eliminating the subdivision of the subparagraphs.

Author Response

Dear Reviewer,

Subject: [Critical Study Quality Management for the Anti-Seepage System in Macau's Landfill Area] Manuscript ID: [applsci-2832056]

We would like to express our sincere gratitude for the time and effort you and the reviewers have dedicated to reviewing our manuscript. We have carefully considered the reviewers' suggestions and have made corresponding revisions to the manuscript. Below, we provide a point-by-point response to the reviewers' comments:

  1. Comment on Introduction Section
    Reviewer's Suggestion: Introduce the discussion in the introduction section differently, followed by the description of Figures 1 and 2.
    Our Response and Action Taken: We have revised the introduction section to reflect this valuable suggestion. The discussion is now introduced in a more coherent and engaging manner. We have also integrated the descriptions of Figures 1 and 2 into this revised narrative, ensuring that they complement the text and provide a clearer understanding of the context.

  2. Comment on Conclusion Section
    Reviewer's Suggestion: Suggest unifying the discussion in the conclusions section by eliminating the subdivision of subparagraphs.
    Our Response and Action Taken: In response to this suggestion, we have restructured the conclusion section. The subsections have been removed to create a more streamlined and cohesive narrative. This change enhances the clarity and impact of our concluding remarks, emphasizing the key findings and implications of our research.

Please find attached the revised manuscript along with a highlighted copy indicating the specific changes made.

We believe that these revisions have significantly improved our manuscript and hope that it is now suitable for publication in [Applied Sciences (ISSN 2076-3417)]. We eagerly await your feedback and are hopeful for a positive decision.

Thank you again for your valuable input and consideration.

Sincerely,